# Exploiting Intermediate Reconstructions in Optical Coherence Tomography for Test-Time Adaption of Medical Image Segmentation

**Thomas Pinetz** [1] (iD)                                        THOMAS.PINETZ@MEDUNIWIEN.AC.AT

**Veit Hucke** [1] (iD)                                                VEIT.HUCKE@MEDUNIWIEN.AC.AT

**Hrvoje Bogunović** [1] (iD)                              HRVOJE.BOGUNOVIC@MEDUNIWIEN.AC.AT

[1] *Institute of Artificial Intelligence, Center for Medical Data Science, Medical University of Vienna, Austria*

**Editors:** Accepted for publication at MIDL 2026

## Abstract

Primary health care frequently relies on low-cost imaging devices, which are commonly used for screening purposes. To ensure accurate diagnosis, these systems depend on advanced reconstruction algorithms designed to approximate the performance of high-quality counterparts. Such algorithms typically employ iterative reconstruction methods that incorporate domain-specific prior knowledge. However, downstream task performance is generally assessed using only the final reconstructed image, thereby disregarding the informative intermediate representations generated throughout the reconstruction process. In this work, we propose IRTTA to exploit these intermediate representations at test-time by adapting the normalization-layer parameters of a frozen downstream network via a modulator network that conditions on the current reconstruction timescale. The modulator network is learned during test-time using an averaged entropy loss across all individual timesteps. Variation among the timestep-wise segmentations additionally provides uncertainty estimates at no extra cost. This approach enhances segmentation performance and enables semantically meaningful uncertainty estimation, all without modifying either the reconstruction process or the downstream model. Code is available here[1].

**Keywords:** Diffusion, Test-Time Adaptation, Uncertainty Estimation, Optical Coherence Tomography

## 1. Introduction

Medical image segmentation is a cornerstone of diagnostic interpretation, with deep learning enabling significant improvements in the robust quantification of biomarkers (Fazekas et al., 2022; Heilemann et al., 2023). Yet, these models are trained primarily on curated, high-fidelity datasets originating from university hospitals (Tripathi et al., 2023). Consequently, their generalization capability often suffers during clinical translation to low-cost imaging hardware (Varoquaux and Cheplygina, 2022).

Recent developments in medical image reconstruction offer a potential solution by enhancing image fidelity and accessibility (McCollough and Rajiah, 2023). While reconstruction was traditionally formulated as an inverse problem solved via iterative optimization

---

1. https://github.com/tpinetz/domain_adaption_by_iterative_reconstruction

of hand-crafted priors (Fei and Luo, 2012; Silva et al., 2010), the paradigm has shifted toward data-driven methods (Hammernik et al., 2016). State-of-the-art approaches now leverage powerful generative frameworks, most notably diffusion (Fazekas et al., 2025; Li et al., 2024), energy-based (Zach et al., 2023), autoregressive (Wang et al., 2023), and flow matching models (Yazdani et al., 2025).

Although these generative models have markedly increased reconstruction quality, they still operate iteratively, refining predictions over multiple steps. Moreover, they follow predetermined time schedule loosely corresponding to the distance from the high-quality domain. Despite this, common evaluation protocols involving downstream tasks, such as biomarker segmentation, rely almost exclusively on the final reconstructed image (Wang et al., 2023; Dong et al., 2025; Tian et al., 2025). While the final reconstructed image is certainly valuable for clinicians, this standard practice neglects the rich information available across the iterative trajectory. Recently, (Jeong et al., 2025) incorporated the segmentation objectives during training of the reconstruction network, however, this requires jointly training both. We hypothesize that standard segmentation networks, even those trained solely on high-quality data, can leverage the intermediate reconstructions of the anatomy to enhance performance without any labeled data. While the visual structure evolves significantly throughout the reconstruction trajectory, we observe strong structural consistency across different subjects at any given timepoint. Coupled with the inherent loss of fine detail during reconstruction, we expect that only minor adaptations to the segmentation network are necessary to handle these time-specific distributions. By analyzing this trajectory of segmentation maps, we aim to provide meaningful semantic uncertainty.

In this work, we propose **I**ntermediate **R**econstruction for **T**est-**T**ime **A**daption (**IRTTA**), a method that modulates an existing segmentation network based on the reconstruction process's time schedule. To this end, we introduce a modulation network that adjusts the downstream model's normalization parameters, given the current timepoint in the reconstruction, and without altering its frozen weights. Addressing the challenges of adaptation without ground truth, we employ a zero-initialization strategy to preserve original performance at the onset and optimize the modulation via entropy minimization (Wang et al., 2021). Crucially, our approach yields multiple predictions per input, enabling uncertainty estimation without requiring additional training of the backbone model.

We further demonstrate the resulting uncertainty estimates and improved segmentation performance on retinal Optical Coherence Tomography (OCT) data (Bogunović et al., 2019) acquired from three different devices, one of which provides substantially higher signal-to-noise ratio (SNR) than the others. Given that OCT segmentation models are known to be sensitive to intensity histogram shifts (Lu et al., 2025), adapting normalization layers is expected to be particularly effective for this modality. Our contributions can be summarized as follows:

- **Novel Modulation Framework:** We propose a method to improve the downstream performance of reconstruction models by exploiting the full reconstruction trajectory.

- **Zero-Shot Uncertainty Estimation:** We provide a mechanism for semantically meaningful uncertainty estimation in pre-trained models without the need for retraining or architectural modification.

- **State-of-the-Art Adaptation:** We achieve superior performance in test-time adaptation for segmentation tasks compared to existing baselines.

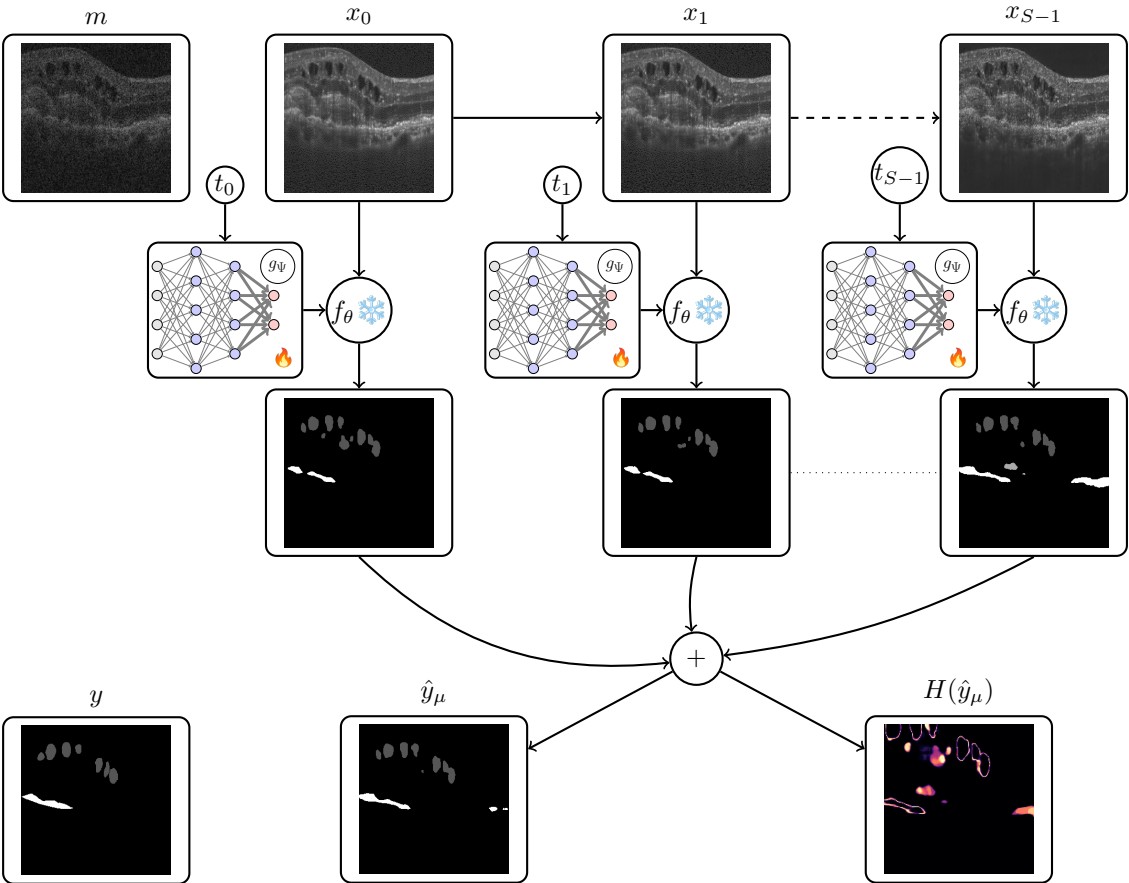

Figure 1: A schematic overview of our proposed method IRTTA. The differences between the reconstructed $x_0, \ldots, x_{S-1}$. Initially, the weights and biases of the final layer in $g_\Psi$ is set to 0, which is highlighted with bold arrows. Hence, the frozen backbone $f_\theta$ retains its performance at initialization and is adapted without labels.

## 2. Method

### 2.1. Problem Formulation

We consider an inverse problem setup where a measurement $m \in \mathbb{R}^M$ is transformed into an image estimate via an iterative reconstruction process. This process yields a sequence of $S$ discrete reconstructions $\mathbf{x} = (x_0, x_1, \ldots, x_{S-1})$, where each $x_i \in \mathbb{R}^d$ represents the estimated image at step $i$, approximating samples from our source domain $\mathcal{D}_\mathcal{S}$.

We assume access to a pre-trained segmentation network $f_\theta : \mathbb{R}^d \to \mathbb{R}^{d \times C}$, parameterized by weights $\theta$, which maps an input image to dense probability maps over $C$ classes.

This network is trained on data from $\mathcal{D}_{\mathcal{S}}$. Our goal is to approximate the ground truth segmentation $y$ by taking advantage of the rich semantic information contained within the entire reconstruction trajectory $\mathbf{x}$, rather than relying solely on the final output $x_{S-1}$.

To achieve this, we introduce a temporal modulation network $g_\Psi(t)$, parameterized by $\Psi$. This network conditions the normalization statistics of $f_\theta$ on the reconstruction time-step $t$. We adopt a test-time adaptation strategy where $g_\Psi$ is optimized via unsupervised entropy minimization, ensuring the segmentation network adapts to the reconstruction dynamics without altering the frozen weights $\theta$. The overall framework is illustrated in Figure 1.

## 2.2. Iterative Reconstruction via Diffusion

While our framework is agnostic to the specific generative model, we formulate our method in the context of diffusion models, which represent the current state-of-the-art in medical image reconstruction (Li et al., 2024; Fazekas et al., 2025).

We utilize a diffusion model $\epsilon_\phi(x, t) : \mathbb{R}^d \times \mathbb{R}^+ \to \mathbb{R}^d$ defined by a time schedule $t \in [0, T]$, discretized into $S$ steps. The reconstruction process generates a trajectory of iterates $\{x_i\}_{i=0}^{S-1}$ corresponding to time points $t_i$. In this context, $t_i$ typically correlates with the distance from the data manifold. Unlike standard generation, in the reconstruction setting, the trajectory is conditioned on the measurement $m$ using a data-consistency term $\mathcal{D}(x, m) :$ $\mathbb{R}^d \times \mathbb{R}^M \to \mathbb{R}$ to ensure fidelity to the measurement (Pinetz et al., 2021; Chung et al., 2023; Li et al., 2024). Otherwise, it follows the diffusion specific update step denoted as Update, which depends on the noise schedule in the chosen model (Karras et al., 2022):

$$\bar{x} = \epsilon_\phi(z_i, t_i)$$
$$x_i = \bar{x} - \tau \nabla \mathcal{D}(\bar{x}, m)$$
$$z_{i+1} \leftarrow \text{Update}(x_i, t_i). \tag{1}$$

The starting point $z_0$ is either pure noise (Chung et al., 2023), the pseudoinverse of the measurement (Li et al., 2024; Fazekas et al., 2025) or a mixture thereof (Gao et al., 2023). We extract the intermediate images $x_i$ after the data-consistency projection, treating them as inputs for the downstream segmentation task.

## 2.3. Test-Time Modulation Network

The general idea is that during the reconstruction the appearance of the images change and small changes are necessary to adapt the segmentation network to perform well along this trajectory. Furthermore, small details might be lost during the reconstruction, which in turn could be useful for the downstream task. For this reason, we adapt the segmentation network $f_\theta$ across the trajectory by injecting time-dependent modulation into its normalization layers (e.g., BatchNorm or LayerNorm). The modulation network $g_\Psi$ accepts the current time-step $t_i$ encoded via sinusoidal embeddings, similar to standard diffusion architectures (Song et al., 2021). The embedding is processed by a Multi-Layer Perceptron (MLP) consisting of two layers with Swish activation. The network $g_\Psi(t_i)$ predicts a set of modulation parameters $(\gamma, \beta)$ for each normalization layer in the backbone $f_\theta$. Let $\bar{x} \in \mathbb{R}^{B \times C \times H \times W}$ denote the output of a standard normalization layer in the frozen network $f_\theta$, defined as:

$$\bar{x} = \bar{\gamma} \cdot \frac{x_{in} - \mu}{\sigma} + \bar{\beta}, \tag{2}$$

where $\mu, \sigma$ are the running statistics and $\bar{\gamma}, \bar{\beta}$ are the frozen affine parameters. We apply the learned modulation as a residual affine transformation:

$$\bar{z} = e^{\gamma} \odot \bar{x} + \beta. \tag{3}$$

Inspired by the zero convolutions used in ControlNet (Zhang et al., 2023b), we also intend for the initialization to retain the performance of the original segmentation network. Therefore, the weight and bias of the final layer of $g_{\Psi}$ is initialized to 0. However, this would change all the predictions to 0 and therefore, we model the scaling factor in log-space ($e^{\gamma}$). This has two distinct advantages:

1. **Sign Stability:** It ensures the scaling factor remains positive, preventing arbitrary sign flips of the features.

2. **Identity Initialization:** By initializing the final projection layer of $g_{\Psi}$ to 0, we obtain $\bar{z} = 1 \cdot \bar{x} + 0$, which reproduces the original pre-trained performance exactly.

### 2.4. Optimization and Uncertainty Estimation

During inference, given a measurement $m$, we perform the reconstruction to obtain pairs $(x_i, t_i)$. We adapt $\Psi$ by minimizing the prediction entropy across the trajectory. Let $\hat{y}_i = \text{Softmax}(f_{\theta, \Psi}(x_i)) \in \mathbb{R}^{d \times C}$ be the soft prediction at step $i$. The unsupervised objective is:

$$\mathcal{L}(\Psi) = -\sum_{i=1}^{S} \frac{1}{d} \sum_{p=1}^{d} \sum_{c=1}^{C} \hat{y}_{i,p,c} \log(\hat{y}_{i,p,c}), \tag{4}$$

where the inner sums compute the spatial entropy of the prediction.

**Inference and Uncertainty:** After adaptation, we compute the ensemble mean prediction $\hat{y}_{\mu} = \frac{1}{S} \sum_{i=1}^{S} \hat{y}_i$. The final semantic segmentation is obtained via $\text{argmax}(\hat{y}_{\mu})$. Furthermore, the pixel-wise entropy of the mean prediction, $H(\hat{y}_{\mu})$, serves as a semantic uncertainty map. As shown in Figure 1, regions of high entropy (bright) correlate with ambiguous anatomical structures.

## 3. Results

In this section, we detail our experimental setup, baselines, and quantitative results. We further provide an ablation study on the modulation architecture and analyze the uncertainty estimation capabilities of our method IRTTA.

### 3.1. Experimental Setup

#### 3.1.1. DATASET AND PREPROCESSING

We utilize the RETOUCH benchmark (Bogunović et al., 2019), which comprises OCT volumes from three different device manufacturers: Cirrus, Topcon, and Spectralis. The Spectralis device is characterized by a notably higher Signal-to-Noise Ratio (SNR) and serves as our reference target domain for high-quality imaging. The dataset includes pixel-wise annotations for fluid-related biomarkers (Intraretinal Fluid (IRF), Subretinal Fluid (SRF),

and Pigment Epithelial Detachment (PED)), which are critical biomarkers for Geographic Atrophy (GA). Due to these properties, RETOUCH is a standard benchmark for domain adaptation in medical imaging (Koch et al., 2022; Gomariz et al., 2025). For standardization, all B-scans were resized to $512 \times 512$ pixels.

### 3.1.2. Implementation Details

We employ a standard U-Net with a ResNet-18 encoder[2] as our downstream segmentation backbone. The network is optimized using Adam with an initial learning rate of $10^{-3}$, a batch size of 8, for 400,000 iterations. We employ cosine annealing to decay the learning rate to $10^{-6}$. Data augmentation includes random 90-degree rotations, shifts, elastic transformations, and contrast adjustments via Albumentations (Buslaev et al., 2020). For the proposed test-time modulation, we freeze the backbone and optimize only the modulation parameters $\Psi$ using Adam with a learning rate of $10^{-5}$. The GARD[3] model was used as our diffusion reconstruction model for Spectralis with number of reconstructions $S = 10$. Hyperparameters were selected via the ablation study detailed in Section 3.4.

Following the RETOUCH protocol (Bogunović et al., 2019), we report the Dice Similarity Coefficient (DSC) evaluated on the full 3D OCT volumes. To avoid biasing the metric towards background predictions (predicting all zeros), evaluation is restricted to cases where fluid is physically present, consistent with standard medical imaging practices.

### 3.2. Baselines and Comparison Methods

We compare our approach against three categories of methods:

1. **General Denoising:** We evaluate SCUNet (Zhang et al., 2023a), assuming that domain shifts in OCT are primarily driven by noise characteristics.

2. **Unsupervised Domain Adaptation (UDA):** We compare with SVDNA (Koch et al., 2022) and SegClr (Gomariz et al., 2025). For these methods, we do 4-fold cross validation to evaluate on the same cases as the other methods. Note that these methods require access to the source domain during training.

3. **Test-Time Adaptation (TTA):** We compare against TENT (Wang et al., 2021), CoTTA (Wang et al., 2022) (both adapting normalization stats), Energy-based adaptation (Zach et al., 2023) and two diffusion based approaches (DDA (Gao et al., 2023) and what we denote as EDM (Chung et al., 2023)). For both methods, we utilized a standard diffusion model implemented via EDM (Karras et al., 2022) trained on the Spectralis dataset, as GARD uses a diffusion process based on gamma distribution (Nachmani et al., 2021) which is non-trivial to adapt.

Additionally, we report an **Oracle** (Supervised) model, where the segmentation network is trained directly on the target domain labels using the same cross validation scheme. We also include a supervised version of our method ($\text{IRTTA}_{sup}$) to quantify the theoretical limit of our architecture.

---

### 3.3. Quantitative Analysis

The results for the Cirrus → Spectralis adaptation task are presented in Table 1. All adaptation methods yield improvements over the baseline. Among the TTA frameworks, our proposed method achieves the highest mean Dice score (0.603), outperforming our baseline GARD (Fazekas et al., 2025) (0.553) and the generic denoiser SCUNet (0.551). GARD was specifically developed for OCT images, and hence already outperforms competing methods.

Notably, SCUNet's strong performance suggests that the domain gap is largely dominated by noise levels. However, our method outperforms SVDNA (Koch et al., 2022), despite SVDNA having access to Cirrus data during training. While our unsupervised approach performs comparably to the best UDA methods, the gap between IRTTA and IRTTA$_{sup}$ (0.603 vs 0.645) indicates that while the normalization adaptation is effective, the unsupervised entropy loss does not fully recover the information available to a supervised signal.

Table 2 presents the results for the Topcon → Spectralis task. Here, our method demonstrates strong generalizability, achieving the highest performance among TTA methods (0.438) using the same hyperparameters as the Cirrus experiments. A qualitative comparison is provided in Figure 2, which highlights the typical differences to the baseline GARD as observed in the dataset. The first row shows an example, where the segmentation is very similar, which is the usual case. In the next two rows, there are subtle changes, where small lesions are added or connected. In the final row an artifact produced by the diffusion process is removed.

Table 1: Comparison of downstream performance on RETOUCH Cirrus → Spectralis (Bogunović et al., 2019). SVDNA and SegClr use Cirrus data during the training phase of the downstream network without labels.

| Method | Venue | DICE | | | |
| --- | --- | --- | --- | --- | --- |
| | | IRF | SRF | PED | Mean |
| Baseline | - | $0.463 \pm 0.29$ | $0.368 \pm 0.24$ | $0.267 \pm 0.27$ | 0.366 |
| SCUNET (Zhang et al., 2023a) | MLR | $0.563 \pm 0.29$ | $\underline{0.617} \pm 0.17$ | $0.474 \pm 0.30$ | 0.551 |
| TENT (Wang et al., 2021) | ICLR | $0.548 \pm 0.21$ | $0.492 \pm 0.25$ | $0.234 \pm 0.23$ | 0.425 |
| CoTTA (Wang et al., 2022) | CVPR | $\mathbf{0.585} \pm 0.18$ | $0.544 \pm 0.23$ | $0.282 \pm 0.24$ | 0.470 |
| Energy (Zach et al., 2023) | TMI | $0.429 \pm 0.29$ | $0.525 \pm 0.20$ | $0.369 \pm 0.28$ | 0.441 |
| DDA (Gao et al., 2023) | CVPR | $0.429 \pm 0.29$ | $0.525 \pm 0.20$ | $0.368 \pm 0.28$ | 0.441 |
| EDM (Chung et al., 2023) | ICLR | $0.281 \pm 0.23$ | $0.378 \pm 0.18$ | $0.476 \pm 0.18$ | 0.378 |
| GARD (Fazekas et al., 2025) | MICCAI | $0.563 \pm 0.23$ | $0.588 \pm 0.21$ | $\underline{0.509} \pm 0.25$ | $\underline{0.553}$ |
| IRTTA | - | $\underline{0.581} \pm 0.23$ | $\mathbf{0.709} \pm 0.15$ | $\mathbf{0.517} \pm 0.27$ | $\mathbf{0.603}$ |
| SVDNA (Koch et al., 2022) | MICCAI | $0.513 \pm 0.29$ | $0.478 \pm 0.28$ | $0.308 \pm 0.32$ | 0.433 |
| SegClr (Gomariz et al., 2025) | MIA | $0.562 \pm 0.26$ | $0.578 \pm 0.24$ | $0.397 \pm 0.29$ | 0.512 |
| IRTTA$_{sup}$ | - | $0.686 \pm 0.21$ | $0.775 \pm 0.09$ | $0.474 \pm 0.30$ | 0.645 |
| Supervised | - | $0.686 \pm 0.20$ | $0.667 \pm 0.21$ | $0.517 \pm 0.32$ | 0.623 |

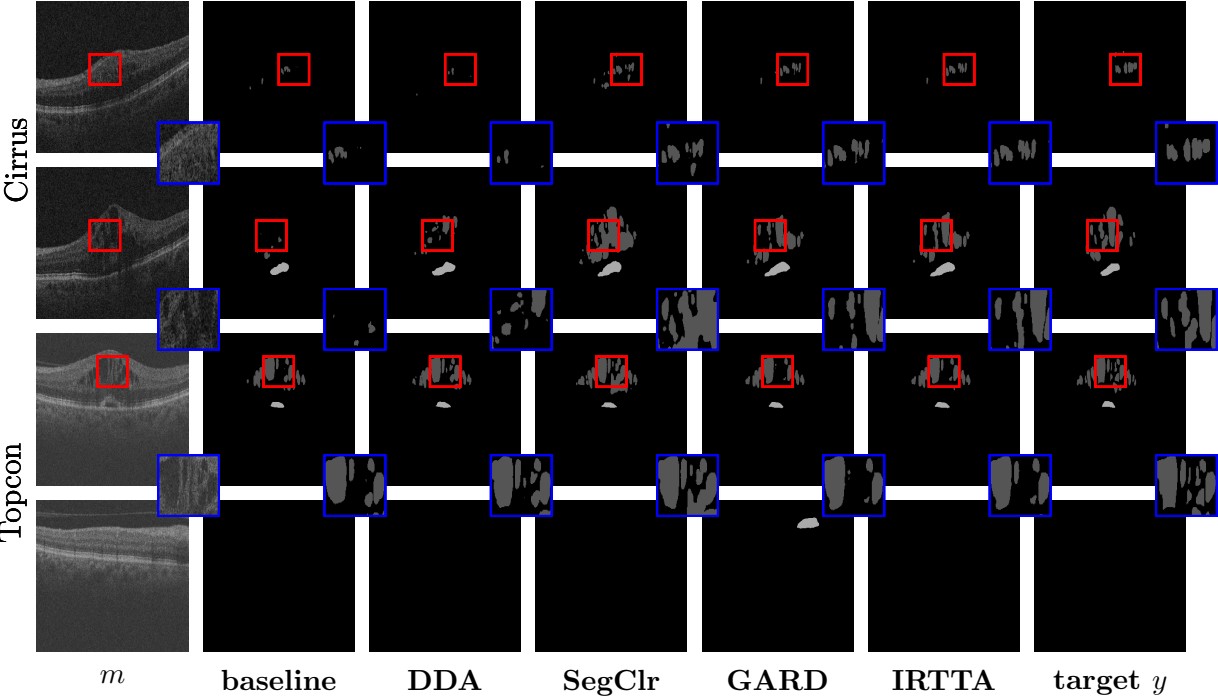

|     | $m$ | baseline | DDA | SegClr | GARD | IRTTA | target $y$ |

Figure 2: Visual comparison of downstream results using different methods. The top and bottom two rows show examples from the Cirrus and Topcon datasets respectively.

### 3.4. Ablation Study

We investigate the impact of key architectural choices on the Cirrus→Spectralis adaptation.

**Trajectory Adaptation:** Table 3 analyzes the benefit of temporal adaptation. "Adapt only last" restricts adaptation to the final reconstructed image ($x_S$), neglecting the trajectory. While this improves over the baseline, our full trajectory method yields superior results (0.603). Similarly "Adapt without first" restricts adaptation to use the trajectory without the first reconstruction as the first step will be the furthest from the target domain and introduce the most noise. However, the final dice score is only slightly lower and therefore not crucial for the adaptation. However, there is still an increase and therefore it is useful to integrate this step into the adaptation. We also compare adapting weights per individual volume ("Per Case") versus sharing adaptation weights across the test set ("Per Dataset"). The negligible performance difference suggests that a single 3D volume contains sufficient statistical diversity to drive the adaptation process effectively.

**Hyperparameter Sensitivity:** Table 4 suggests that the embedding size of the modulation network is not a critical hyperparameter, though performance degrades with excessively large embeddings ($> 64$), likely due to overfitting the limited modulation signal. Conversely, increasing the number of adaptation steps initially yields minor performance gains (Table 5). These improvements saturate at 100 steps, after which performance deteriorates. Similarly, increasing the number of reconstructions improves the performance and saturates at $S = 10$) and drops slightly with $S = 20$ (see Table 6). Furthermore, we added

Table 2: Comparison of downstream performance on RETOUCH Topcon → Spectralis (Bogunović et al., 2019). SVDNA and SegClr use Topcon data during the training phase of the downstream network without labels.

| Method | Venue | DICE | | | |
|--------|-------|------|------|------|------|
| | | IRF | SRF | PED | Mean |
| Baseline | - | $0.475 \pm 0.27$ | $0.282 \pm 0.21$ | $0.398 \pm 0.16$ | 0.385 |
| SCUNET (Zhang et al., 2023a) | MLR | $0.446 \pm 0.26$ | $0.213 \pm 0.18$ | $\mathbf{0.518} \pm 0.14$ | 0.393 |
| DDA (Gao et al., 2023) | CVPR | $0.461 \pm 0.27$ | $0.362 \pm 0.27$ | $0.384 \pm 0.22$ | 0.402 |
| EDM (Chung et al., 2023) | ICLR | $0.294 \pm 0.24$ | $0.326 \pm 0.19$ | $0.504 \pm 0.17$ | 0.375 |
| Energy (Zach et al., 2023) | TMI | $0.464 \pm 0.26$ | $0.241 \pm 0.16$ | $0.478 \pm 0.14$ | 0.394 |
| TENT (Wang et al., 2021) | ICLR | $\underline{0.553} \pm 0.25$ | $0.362 \pm 0.29$ | $0.289 \pm 0.22$ | 0.401 |
| CoTTA (Wang et al., 2022) | CVPR | $\mathbf{0.567} \pm 0.25$ | $\mathbf{0.400} \pm 0.30$ | $0.302 \pm 0.22$ | $\underline{0.423}$ |
| GARD (Fazekas et al., 2025) | MICCAI | $0.502 \pm 0.26$ | $0.271 \pm 0.16$ | $\underline{0.488} \pm 0.15$ | 0.420 |
| IRTTA | - | $0.507 \pm 0.27$ | $\underline{0.377} \pm 0.25$ | $0.447 \pm 0.15$ | $\mathbf{0.444}$ |
| SVDNA (Koch et al., 2022) | MICCAI | $0.488 \pm 0.25$ | $0.419 \pm 0.26$ | $0.438 \pm 0.20$ | 0.448 |
| SegClr (Gomariz et al., 2025) | MIA | $0.582 \pm 0.24$ | $0.487 \pm 0.28$ | $0.483 \pm 0.20$ | 0.517 |
| IRTTA$_{sup}$ | - | $0.526 \pm 0.27$ | $0.353 \pm 0.27$ | $0.534 \pm 0.14$ | 0.471 |
| Supervised | - | $0.625 \pm 0.22$ | $0.481 \pm 0.29$ | $0.521 \pm 0.20$ | 0.542 |

run time statistics for our approach with increasing reconstruction steps. The training itself is not a large bottleneck.

Table 3: Ablation: Trajectory vs. Single-step adaptation (Cirrus).

| Method | DICE | | | |
|--------|------|------|------|------|
| | IRF | SRF | PED | Mean |
| Adapt only last | $0.560 \pm 0.24$ | $0.678 \pm 0.17$ | $0.507 \pm 0.26$ | 0.581 |
| Adapt without first | $0.575 \pm 0.23$ | $0.697 \pm 0.16$ | $\mathbf{0.519} \pm 0.26$ | 0.597 |
| Adapt per case | $0.570 \pm 0.23$ | $\mathbf{0.711} \pm 0.15$ | $0.513 \pm 0.28$ | $\mathbf{0.603}$ |
| Adapt per dataset | $\mathbf{0.581} \pm 0.23$ | $0.709 \pm 0.15$ | $0.517 \pm 0.27$ | $\mathbf{0.603}$ |

### 3.5. Uncertainty visualization

We quantify the reliability of our uncertainty estimates using the Expected Calibration Error (ECE) and Precision-Recall Area Under the Curve (PRAUC), presented in Table 7. The PRAUC is computed in a binary classification setting, where the probabilities of the fluid classes are summed up after softmax. Our method reduces the ECE from $\sim 0.007$ vs $\sim 0.013$ and $\sim 0.003$ vs $\sim 0.005$ compared to the baseline GARD, indicating that our approach aligns the confidence scores better with the true accuracy. Similarly, the PRAUC values are improve from 0.532 to 0.672 on Cirrus and from 0.447 to 0.454 on Topcon. For the PRAUC, the difference to the supervised trained model is more pronounced. Qualita-

Table 4: Ablation: Size of the embedding vector (Cirrus).

| Emb size | DICE | | | |
|---|---|---|---|---|
| | **IRF** | **SRF** | **PED** | **Mean** |
| 4 | $0.579 \pm 0.23$ | $0.697 \pm 0.16$ | $\mathbf{0.531} \pm 0.26$ | 0.602 |
| 8 | $0.578 \pm 0.23$ | $0.704 \pm 0.16$ | $0.526 \pm 0.26$ | **0.603** |
| 16 | $\mathbf{0.581} \pm 0.23$ | $\mathbf{0.709} \pm 0.15$ | $0.517 \pm 0.27$ | **0.603** |
| 32 | $0.572 \pm 0.24$ | $0.709 \pm 0.15$ | $0.516 \pm 0.27$ | 0.599 |
| 64 | $0.550 \pm 0.25$ | $0.705 \pm 0.13$ | $0.480 \pm 0.29$ | 0.578 |
| 128 | $0.496 \pm 0.26$ | $0.679 \pm 0.15$ | $0.434 \pm 0.32$ | 0.536 |

Table 5: Ablation: Number of steps used in the adaption (Cirrus).

| Steps | DICE | | | |
|---|---|---|---|---|
| | **IRF** | **SRF** | **PED** | **Mean** |
| 1 | $0.579 \pm 0.23$ | $0.670 \pm 0.19$ | $0.541 \pm 0.26$ | 0.597 |
| 10 | $0.580 \pm 0.23$ | $0.674 \pm 0.18$ | $\mathbf{0.541} \pm 0.26$ | 0.598 |
| 50 | $0.580 \pm 0.23$ | $0.691 \pm 0.17$ | $0.536 \pm 0.26$ | 0.602 |
| 100 | $\mathbf{0.581} \pm 0.23$ | $\mathbf{0.709} \pm 0.15$ | $0.517 \pm 0.27$ | **0.603** |
| 500 | $0.465 \pm 0.26$ | $0.613 \pm 0.20$ | $0.386 \pm 0.33$ | 0.488 |

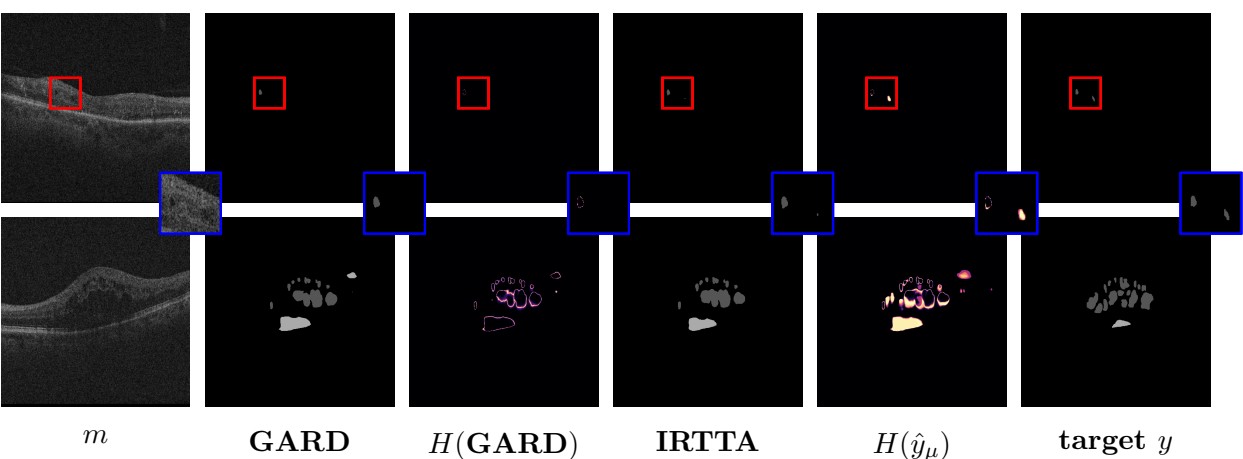

$m$  **GARD**  $H(\mathbf{GARD})$  **IRTTA**  $H(\hat{y}_\mu)$  **target** $y$

Figure 3: Visual comparison of uncertainty estimation compared to the baseline, visualized by showing the entropy of the prediction. The uncertainty now shows semantically meaningful information instead of the boundary of the initial segmentation.

tively (Figure 3), our entropy maps do not merely highlight object boundaries but correctly identify ambiguous anatomical regions, such as potential lesions that exist in the ground truth but are degraded in the input, offering valuable interpretability for clinicians. This is especially striking in the first row, where a small lesion is lost after the reconstruction

Table 6: Ablation: Number of reconstructions used in the adaption (Cirrus).

| S | DICE | | | | Run Time |
|---|---|---|---|---|---|
| | IRF | SRF | PED | Mean | in s |
| 5 | $0.539 \pm 0.23$ | $0.648 \pm 0.18$ | $0.497 \pm 0.26$ | 0.561 | 16 |
| 10 | $0.580 \pm 0.23$ | $0.674 \pm 0.18$ | $\mathbf{0.541} \pm 0.26$ | 0.598 | 69 |
| 15 | $0.593 \pm 0.23$ | $\mathbf{0.720} \pm 0.13$ | $0.500 \pm 0.30$ | **0.605** | 121 |
| 20 | $\mathbf{0.604} \pm 0.20$ | $0.695 \pm 0.13$ | $0.474 \pm 0.32$ | 0.591 | 196 |

Table 7: Uncertainty Quantification (Expected Calibration Error and Precision-Recall Area Under the Curve).

| Method | ECE | | PRAUC | |
|---|---|---|---|---|
| | Cirrus | Topcon | Cirrus | Topcon |
| GARD | $.01332 \pm .010$ | $.00455 \pm .004$ | $0.532 \pm 0.26$ | $0.447 \pm 0.24$ |
| IRTTA | $.00697 \pm .007$ | $.00342 \pm .005$ | $0.672 \pm 0.22$ | $0.454 \pm 0.24$ |
| Supervised | $.00534 \pm .005$ | $.00363 \pm .004$ | $0.765 \pm 0.21$ | $0.614 \pm 0.25$ |

process. However this lesion is found by the segmentation network at some point in the trajectory and hence visualized in the uncertainty map.

## 4. Conclusion

In this work, we demonstrated that the iterative nature of modern reconstruction algorithms offers a rich, yet underutilized, source of semantic information. By modulating a pre-trained segmentation network based on the reconstruction trajectory, we achieved significant performance gains on out-of-distribution data via ad-hoc test-time adaptation.

Crucially, our method achieves performance competitive with dedicated Unsupervised Domain Adaptation (UDA) frameworks, despite the distinct advantage of not requiring access to the source domain during training. While a performance gap remains compared to fully supervised upper bounds, our ablation studies suggest that future improvements may stem from more sophisticated fine-tuning strategies rather than loss function engineering alone. However, the performance deterioration after more than hundred of iterations suggests that an improved loss function might increase the stability and therefore improve the robustness of the hyperparameters.

Future research will focus on validating this approach across different imaging modalities, such as MRI/CT reconstruction. In that space multiple iterative reconstruction models exist to be tested (Safari et al., 2026) as well as datasets with either paired devices (Islam et al., 2025) or including downstream tasks (Zbontar et al., 2018). Additionally, we aim at exploring fusion mechanisms to replace naive ensemble averaging, thereby maximizing the utility of the intermediate representations.

## Acknowledgments

Funded by the European Union, EIC-2023-PATHFINDEROPEN-01 (I-SCREEN, grant no. 101130093). Views and opinions expressed are however those of the author(s) only and do not necessarily reflect those of the European Union or European Innovation Council and SMEs Executive Agency (EISMEA). Neither the European Union nor the granting authority can be held responsible for them.

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
