# OpenReview forum: "Exploiting Intermediate Reconstructions in Optical Coherence Tomography for Test-Time Adaption of Medical Image Segmentation"
_MIDL.io/2026/Conference — MIDL 2026 Poster_

### Official Review · Reviewer_9YV7 · 2025-12-26

**Confidence:** 4
**Preliminary Rating:** 4
**Final Rating:** 4

**Summary:**

The authors introduce a test-time adaptation method that leverages intermediate OCT reconstructions to adapt a frozen downstream segmentation network via time-conditioned normalization. They show that this improves cross-device segmentation performance and enables meaningful uncertainty estimation without retraining or having access to the source-domain. The results on the RETOUCH benchmark dataset demonstrate consistent gains over existing test-time adaptation baselines

**Strengths:**

The core idea of the proposed method of using intermediate steps of an iterative reconstruction process that may contain useful semantic information for segmentation is novel and relevant. The approach leaves both the reconstruction model and segmentation network unchanged and this modularity is a strong point that could make deployment easier. The estimation of uncertainty maps is elegant and seems to add relevant information.

**Weaknesses:**

The proposed method assumes access to all intermediate reconstructions. While reasonable in research settings, this may limit its applicability to proprietary scanners or retrospective datasets for which only the final images were saved. The results are convincing, but the Dice coefficient gains over strong baselines remain minimal in certain settings. Although the RETOUCH datasets includes different device manufactures, this is the only dataset considered in the study.

**Detailed Comments:**

-It is not clear whether the improvements come mostly from the noisy first steps.

-Including a discussion or evidence for applicability of the proposed method to other modalities (e.g., MRI, CT) or non-diffusion iterative solvers would strengthen the authors' claims.

-Figure 1: zoomed-in images could be shown instead as it is difficult to see the differences.

**Justification Of Final Rating:**

The authors have performed additional experiments in the rebuttal phase, answering to all my concerns. This is an interesting work and could be of great interest to researchers in the medical imaging field.

**Justification Of The Preliminary Rating:**

This paper presents a well-motivated and technically sound approach to test-time adaptation by exploiting intermediate reconstruction trajectories, yielding improved OCT segmentation and meaningful uncertainty estimates. It can be of great interest to researchers in the field.

**Questions To Address In The Rebuttal:**

-Running the segmentation backbone for $S=100$ reconstruction steps likely consists in a significant computational bottleneck. Can the authors comment on that and report the computational cost and time differences?

-There is a considerable drop in performance when increasing the number of steps to 500 that hints to a possibly unstable loss function. How do the authors explain these results?

---

> ### Author Response · Authors · 2026-01-22
> **Rebuttal**
>
> > It is not clear whether the improvements come mostly from the noisy first steps.
>
> We address this concern with a new experiment, labeled 'Adapt without first' in Table 3.
> The results demonstrate that omitting the initial step leads to only marginal performance degradation
>
> > Including a discussion or evidence for applicability of the proposed method to other modalities (e.g., MRI, CT) or non-diffusion iterative solvers would strengthen the authors' claims.
>
> We agree that more modalities would strengthen the paper and in principle the pipeline does not exploit any OCT specific characteristics and should work out of the box on other modalities and reconstruction models. However, this is a large amount of work as the data handling, downstream task and baseline comparisons change. This is not feasible in the rebuttal period. However, we have plans to do this and updated the discussion to point to relevant modalities and reconstruction models to try as a next step.
>
> > zoomed-in images could be shown instead as it is difficult to see the differences.
>
> We adapted the figure to show zooms accordingly.
>
> > Running the segmentation backbone for $S=100$ reconstruction steps likely consists in a significant computational bottleneck. Can the authors comment on that and report the computational cost and time differences?
>
> We apologize for the ambiguity in our initial description. To clarify: our original ablation studied the number of *optimization steps* used to adapt the modulation network, while the number of reconstruction steps $S$ was fixed to $10$ (following the GARD protocol). We have rewritten this section to clearly distinguish between these two parameters.
>
> Furthermore, to fully address your concern, we added a new ablation on the number of reconstruction steps $S$ and a runtime analysis in Table 6.
>
> As expected, the total inference time scales linearly with $S$, as the overhead from the modulation adaptation is negligible. Specifically, the total time is approximately the forward pass time of the network $T$ multiplied by $S$. The adaptation, with $S=10$ using $100$ optimization steps requires $69$ seconds on a single NVIDIA A6000 GPU.
>
> We have updated the manuscript to clarify the experimental setup and included the new runtime analysis in Table 6.
>
> > There is a considerable drop in performance when increasing the number of steps to 500 that hints to a possibly unstable loss function. How do the authors explain these results?
>
> We acknowledge the reviewer's concern regarding optimization stability. However, we note that instability is a pervasive challenge in unsupervised test-time adaptation, often caused by error accumulation in the absence of ground truth. This is why established methods like TENT restrict updates solely to normalization parameters and often require episodic resets.
>
> To investigate this further, we evaluated the CoTTA baseline (as suggested by Reviewer #2) and observed similar performance deterioration after $50$ update steps. Crucially, when we repeat our experiment using ground truth labels and a cross-entropy loss (Ours$_{sup}$), this instability vanishes, and the model trains robustly for thousands of iterations.
>
> This confirms that the limitation lies in the unsupervised nature of the objective function rather than the modulation architecture itself. Developing loss functions that remain stable over long-term continuous adaptation is an open research question.
>
> We have updated the Conclusion to suggest directions for future work on robust unsupervised objectives.

---

> > ### Comment · Reviewer_9YV7 · 2026-01-29
> >
> > Thank you for addressing my concerns.

---

### Official Review · Reviewer_zAyR · 2026-01-09

**Confidence:** 3
**Preliminary Rating:** 3
**Final Rating:** 4

**Summary:**

This paper proposes a test-time adaptation method that modulates an existing segmentation network based on the reconstruction process’s time schedule. The modulation employs a zero-initialization strategy and is optimized via entropy minimization. It also enables uncertainty estimation with the multiple predictions per input. Experiments are conducted on Optical Coherence Tomography data and show superior performance.

**Strengths:**

1. Paper is very well written. Problem formation is clear, method is well explained and experiments are carefully conducted. Code is available on GitHub which makes the paper easy to reproduce.
2. The approach is novel and has several practical advantages:
1) method is agnostic to generative reconstruction models
2) no source domain data is needed
3) uncertainty estimation comes for free
3. Experiments are comprehensive. Multiple baselines including denoising, UDA and TTA are compared, and ablation studies are thorough.

**Weaknesses:**

1. The experiments are done on only one modality and one reconstruction model, which weakens the claim of generalizability to other modalities and reconstruction methods.
2. The uncertainty estimation evaluation should benefit from adding some AUROC analysis to see whether there are failure predictions.

**Detailed Comments:**

1. What’s the motivation/reference of injecting time-dependent modulation into normalization layers of the generative network? And why is it supposed to work?
2. What’s the additional computational cost of running segmentation across all time steps?
3. How does the proposed method compare with other TTA methods? For example, https://arxiv.org/abs/2203.13591

**Justification Of Final Rating:**

The authors have made several changes regarding to my comments. Some of the questions should be done in the initial submission to make the paper stronger. But given the new added motivation and analysis, I raise my score to 4.

**Justification Of The Preliminary Rating:**

The method is novel and well presented within its scope, but it could be better with more justification of the effectiveness in terms of 1) motivation, 2) experiment across modalities and reconstruction models, 3) stronger results.

**Questions To Address In The Rebuttal:**

See above.

---

> ### Author Response · Authors · 2026-01-22
> **Rebuttal**
>
> > The uncertainty estimation evaluation should benefit from adding some AUROC analysis to see whether there are failure predictions.
>
> Given the high class imbalance in our dataset (where anomalous pixels constitute less than 1%), we adopted the Area Under the Precision-Recall Curve (AUPRC) as a more reliable performance metric. We have included these results in Table 7 and updated the section to elaborate on this choice.
>
>
> > What’s the motivation/reference of injecting time-dependent modulation into normalization layers of the generative network? And why is it supposed to work?
>
> Unlike standard TTA approaches (e.g., TENT[1], CoTTA[2]) that perform static adaptation of normalization parameters, our approach leverages the temporal dynamics of the diffusion process to modulate the normalization layers of the *segmentation* network.
>
> We observed that the domain gap is not static; rather, reconstructions at different timepoints $t_i$ exhibit distinct but consistent visual features. Furthermore, the reconstruction process inherently filters out certain high-frequency details. While this filtering often removes noise, it can occasionally remove small semantic objects.
>
> A key motivation for our approach is illustrated in Figure 3 (Row 1): a small lesion is preserved at intermediate timesteps but is smoothed out in the final reconstruction. By injecting time-dependent modulation via a hypernetwork, our model adapts to the changing feature distribution at each step $t_i$. Consequently, even if the lesion is absent from the final hard segmentation, the model successfully captures its presence in the uncertainty map. This demonstrates that adapting to the full trajectory yields a more robust representation than relying solely on the final output.
>
> We have updated the manuscript to emphasize these observations.
>
>
> > What’s the additional computational cost of running segmentation across all time steps?
>
> Typically the cost increases linearly with the reconstruction points considered as the overhead of the modulation network is minimal. The training time for the modulation network for $S=10$ and 100 steps requires 69 seconds on a single A6000 GPU. This is marked in Table 6.
>
> > The experiments are done on only one modality and one reconstruction model, which weakens the claim of generalizability to other modalities and reconstruction methods.
>
> We agree that more modalities would strengthen the paper and in principle the pipeline does not exploit any OCT specific characteristics and should work out of the box on other modalities. However, this is a large amount of work as the data handling, downstream task and baseline comparisons change. This is not feasible in the rebuttal period. However, we have plans to do this and updated the discussion to point to relevant modalities and reconstruction models to try as a next step.
>
>
> > How does the proposed method compare with other TTA methods? For example, https://arxiv.org/abs/2203.13591
>
> We included a comparison to CoTTA, where we tried the same number of update steps as in our ablation study and choose the best performing one, and we compare favorably to it on both devices (13.3 and 2.1 more average dice on Cirrus and Topcon respectively).
>
> [1] Wang et al. "Tent: Fully test-time adaptation by entropy minimization." ICLR 2021
>
> [2] Wang et al. "Continual test-time domain adaptation" CVPR 2022

---

> ### Author Response · Authors · 2026-02-02
> **Rating**
>
> Thank you for your comment and nice words! Would you consider to also update the final rating such that it matches your updated preliminary one and your comment?

---

### Official Review · Reviewer_85yz · 2026-01-11

**Confidence:** 5
**Preliminary Rating:** 5
**Final Rating:** 5

**Summary:**

This work introduces a novel test-time adaptation framework that leverages the intermediate reconstructions produced by iterative generative models, such as diffusion-based approaches, in Optical Coherence Tomography (OCT). Instead of relying solely on the final reconstructed image, the method modulates normalization layers of a frozen segmentation network using a lightweight temporal modulation network conditioned on reconstruction timesteps. Ablation studies confirm the importance of trajectory-wide adaptation and show that uncertainty estimates derived from timestep variation are both semantically meaningful and well-calibrated. The significance lies in bridging reconstruction and downstream analysis, offering a practical, zero-shot strategy to enhance clinical robustness of medical image segmentation without retraining or altering backbone architectures.

**Strengths:**

The paper introduces the idea of exploiting intermediate reconstructions from iterative generative models for test-time adaptation. This is a fresh perspective that bridges reconstruction and downstream segmentation, offering a new dimension of information that is typically discarded. By modulating normalization layers of a frozen segmentation network, the method avoids retraining or architectural changes, making it lightweight and directly applicable in clinical workflows where retraining is often infeasible.
The framework provides semantically meaningful uncertainty estimates at no extra cost, which is highly valuable in medical imaging where confidence calibration is critical for clinical decision-making.

**Weaknesses:**

The approach assumes access to intermediate reconstructions from iterative generative models. This limits applicability to settings where such trajectories are available, and may not generalize to non-iterative or single-pass reconstruction methods.
While the manuscript is generally well-written, some sections (particularly the Methods) are dense with technical detail and could benefit from more intuitive explanations or schematic illustrations to aid readers outside the immediate subfield.
These weaknesses do not undermine the novelty of the contribution but highlight areas where the work could be strengthened.

**Detailed Comments:**

The explanation of why modulation parameters are modeled in log-space is incomplete. Expanding this rationale with either theoretical intuition or empirical evidence would improve transparency.

**Justification Of Final Rating:**

This paper makes a novel and valuable contribution by introducing a test-time adaptation framework that leverages intermediate reconstructions from iterative generative models, a source of information that is typically ignored in downstream tasks. The approach modulates normalization layers of a frozen segmentation network without retraining, making it lightweight and directly applicable in clinical workflows.

**Justification Of The Preliminary Rating:**

This paper makes a novel and valuable contribution by introducing a test-time adaptation framework that leverages intermediate reconstructions from iterative generative models, a source of information that is typically ignored in downstream tasks. The approach modulates normalization layers of a frozen segmentation network without retraining, making it lightweight and directly applicable in clinical workflows.

**Questions To Address In The Rebuttal:**

.

---

> ### Author Response · Authors · 2026-01-22
> **Rebuttal**
>
> Thank you for your encouraging review! We updated the manuscript to improve the motivation and clarity of our paper.
>
> > The explanation of why modulation parameters are modeled in log-space is incomplete. Expanding this rationale with either theoretical intuition or empirical evidence would improve transparency.
>
> Our intuition comes from the ControlNet paper[1], where zero convolutions are used to adapt another generation network. The weights and biases of these convolutions are initialized to $0$ and hence the control net does not change the original network.
>
> We do the same with our final projection layers, which are also initialized to $0$. However, because we are adapting the normalization layers and not the features directly, we have to adapt the output in a way such that the initialization is retained.
> As a remainder, the modulation parameters are $\beta, \gamma$ and the modulation is performed as follows: $\bar z = \exp(\gamma)\odot \bar{x} + \beta$, where $\bar{x}$ is the normalized output of the original normalization layer.
>
> One way to retain the initialization is $\exp(\gamma)$, but there are other methods as well, such as $1 + \gamma$. In initial experiments, we tried $1 + \gamma$ and the performance was only marginally worse. We hypothesized that changing the sign of $\gamma$ in the middle of the network would lead to instabilities and should henceforth be avoided; for that, $\exp$ is well suited. However, there are still multiple options such as $1 + \gamma^2$ or $\max(1 + \gamma, 0)$. However, we conjecture that the performance differences between these choices will be minimal and so we stuck with the exponential one.
>
> We updated the teaser figure MLPs to reflect the $0$ initialization and updated the manuscript to better explain this section.
>
>
> [1] Zhang, Lvmin, Anyi Rao, and Maneesh Agrawala. "Adding conditional control to text-to-image diffusion models." CVPR. 2023.

---

### Author Rebuttal · Authors · 2026-01-22

**Rebuttal:**

We thank all of the reviewers for their valuable feedback and we are especially delighted that all of them highlighted the novelty of our work.

We have updated the manuscript in the following ways:

* **New Baseline:** We added a new baseline, CoTTA.
* **New Ablation:** Adapting without the first reconstruction $x_0$ (Table 3).
* **New Ablation:** Number of reconstructions $S$ and runtime (Table 6).
* **New Evaluation:** PRAUC results included in Table 7.
* **Figures:** Updated teaser figure with zooms.
* **Readability:** Updated manuscript to improve readability and add motivation.

New sections are highlighted in **red**.

**Supporting Material:**

/attachment/c6a4682a3bbec3ced62b6c8817ec1dcf203b9153.pdf

---

### Meta-Review · Area_Chair_EXqs · 2026-02-09

**Recommendation:** Accept (Oral)
**Confidence:** 3

**Metareview:**

The reviewers agree that the method is novel and appreciated the additional experiments provided by the authors.

---

### Decision · Program_Chairs · 2026-02-13

Accept (Poster)